# The Emergency Performance of the Hungarian Ambulance Service during the COVID-19 Pandemic

**DOI:** 10.3390/healthcare10112331

**Published:** 2022-11-21

**Authors:** Klára Bíró, Máté Sándor Deák, György Pápai, Attila Nagy, Viktor Dombrádi, Gábor Tamás Szabó, Klára Boruzs, Gábor Bányai, Gábor Csató

**Affiliations:** 1Institute of Health Economics and Management, Faculty of Economics and Business, University of Debrecen, 4032 Debrecen, Hungary; 2Doctoral School of Health Sciences, University of Debrecen, 4028 Debrecen, Hungary; 3Hungarian National Ambulance Service, 1055 Budapest, Hungary; 4Department of Health Informatics, Institute of Health Sciences, Faculty of Health Sciences, University of Debrecen, 4028 Debrecen, Hungary; 5Health Services Management Training Centre, Faculty of Health and Public Administration, Semmelweis University, 1125 Budapest, Hungary; 6Division of Cardiology, Department of Cardiology, Faculty of Medicine, University of Debrecen, 4032 Debrecen, Hungary

**Keywords:** COVID-19, pandemic, acute myocardial infarction, stroke, ambulance service, Hungary

## Abstract

The COVID-19 pandemic had a considerable impact on the whole health sector, particularly on emergency services. Our aim was to examine the performance of the Hungarian National Ambulance Service during the first four waves of the pandemic. We defined the 2019 performance of the service as the baseline and compared it with the activity during the pandemic years of 2020 and 2021. The data contained deliveries related to acute myocardial infarction, hemorrhagic stroke, ischemic stroke, overall non-COVID-related ambulance deliveries, COVID screenings performed by the ambulance service, and COVID-related ambulance deliveries. The data were aggregated for each week of the investigated time period and stratified by gender and age. Compared with the pre-pandemic era, we found a significant increase in all three medical conditions and overall deliveries (*p* < 0.001 in all cases). As a result of the increased burden, it is important for emergency services to prepare for the next global epidemic and to improve organizational performance and rescue activities. The Hungarian example highlights that in a pandemic, it can be beneficial to organize the emergency care of a country or a larger region under a single provider with a single decision maker supported by business intelligence.

## 1. Introduction

The coronavirus disease 2019 (COVID-19) pandemic is an unparalleled challenge for the health sector, mostly for pulmonology departments, followed closely by emergency care departments and ambulance services [1,2,3,4,5,6,7]. Soon after the appearance of this novel virus, it caused a considerable burden for healthcare systems globally. In Hungary, the first case was reported on 4 March 2020 [8]. The first reported COVID-19-related death occurred on 15 March [9]. On 18 March, the Operational Corps Responsible for the Containment of the Coronavirus Epidemic (Koronavírus-járvány Elleni Védekezésért Felelős Operatív Törzs) declared that the infectious virus could be present anywhere in Hungary [10]. Hungary has since experienced its fourth COVID-19 wave, resulting in the deaths of more than 48,000 citizens [11].

In Hungary, the National Ambulance Service (NAS) is responsible for all ambulance-related tasks nationwide, and during the pandemic, their role became even more important. In addition to the general care, in late March 2020, the service received additional duties. In cooperation with the universities of clinical medicine, the screening of possible COVID-19 patients became the task of the NAS, and the transportation of all confirmed cases of patients was also their responsibility [12].

The NAS already faced considerable challenges before the pandemic because the Hungarian health indicators are among the worst in the European Union [13,14,15]. Therefore, it is imperative to understand the degree to which the COVID-19 wave increased the workload of the “first responders” in health care. Despite the importance of the topic, only one study investigated how emergency ambulance deliveries within the United Kingdom were affected by the COVID-19 pandemic [16]. Because this study was conducted relatively early during the pandemic, the researchers could only investigate the first COVID-19 wave; thus, there is no information regarding the impact of the following more severe waves, which could be useful for those organizing emergency ambulance services. Although other articles have discussed the increased burden on ambulance services during the COVID-19 pandemic, these published papers mainly focused on how to identify COVID-19 infections and other factors related to medical technology [17,18,19,20]. Owing to the lack of knowledge and the concern that a new global pandemic may emerge in the future, it is important to investigate this question thoroughly. Therefore, the primary aim of this research was to examine the performance of the NAS during the first four waves of the pandemic.

## 2. Materials and Methods

This is a retrospective follow-up study of Hungarian citizens conducted to estimate the incidence of cases of emergency assistance in the period of 2019–2021. All data regarding ambulance activities were provided by the management of the NAS, which is the largest medical and ambulance institution in Hungary, and which operates the national registry of ambulance performance implemented in late 2018. In order to obtain at least two whole years of baseline data before the COVID-19 pandemic, the original intention was to obtain data starting from January 2018. However, because the NAS only implemented a uniform data recording method in late 2018, the time period under consideration ranged from January 2019 to December 2021. The subjects of our study are the deliveries and COVID-19 screenings related to patients aged 15 years and older and performed by the NAS from 2019 to 2021. Our outcome variables were annual number of deliveries of patients with diagnosis of COVID-19 (ICD-10: U07.1), COVID-19 screening (ICD-10: Z11.52), acute myocardial infarction (ICD10: I21, I22), haemorrhagic stroke (ICD10: I60, I61, I62), ischemic stroke (ICD10: I63, I64), and non-COVID-19 (ICD-10: all but U07.1 and Z11.52). The specific medical conditions were chosen because, in connection with ambulance deliveries, it is the most important for these three diseases to start with the appropriate quality of care at the right time. These data were aggregated for each week of the investigated time period, then stratified by gender and age (15–64 vs. 65+).

For the descriptive statistical analysis ratios, means, standard deviations, medians, and interquartile ranges were calculated for each subgroup. The Shapiro–Wilk test was used to investigate the distribution of the executed COVID-19 screenings and ambulance activities regarding the aforementioned types of deliveries.

To determine whether the distribution of gender and age significantly differed between the three investigated years, Pearson’s chi-squared test was used. Cases in which gender or age were not recorded by the ambulance staff were excluded from the analysis. Student’s *t*-test was applied to explore whether the number of COVID-19 screenings and the number of investigated ambulance deliveries significantly changed during the period of 2020–2021.

A weekly moving average was calculated to visualize the overall trends of acute myocardial infarction, stroke, and non-COVID-19-related ambulance deliveries. In addition, we calculated the upper and lower 95% standard deviations using the year 2019 as a baseline reference. This approach was adopted from a British study investigating emergency ambulance deliveries during the first COVID-19 wave for STEMI (ST elevation myocardial infarction) and stroke [16]. By doing so, the goal was to explore which weeks had considerably more or fewer deliveries. Additionally, a linear trend was inserted into each figure to visualize the overall trends. R-squared values were calculated to determine how these trends fit the dataset. These calculations were not conducted for COVID-19 screenings and COVID-19 -elated deliveries, as there were no such cases in 2019.

To show how the screenings and deliveries are related to the four COVID-19 waves, we calculated the beginning and end of each wave in Hungary. There is an ongoing debate on how to define the start and the end of any given COVID-19 wave [21,22]. Two methods were considered by our research team. We determined that adopting the method used in Chile might be subject to limitations in Hungary, as this approach assumes that the two countries have a similar population density [21]. Therefore, we opted to use the method that considers the reproduction rate (R) of the virus [22]. Based on this approach—with the exception of the very first case identified in Hungary—the start of a COVID-19 wave was defined by an R value of more than 1 for two consecutive weeks, and the end of a COVID-19 wave was marked two weeks before the R value increased to more than 1. The R values regarding the COVID-19 pandemic in Hungary were obtained from the Our World in Data website [23]. As the earliest data were from 22 March 2022, we decided that the first confirmed COVID-19 case in Hungary would be the start of the first COVID-19 wave [8].

Microsoft Excel was used for the descriptive statistical analysis and to create the tables and figures. Intercooled STATA v13 was used to evaluate the normality of continuous variables and for all comparative statistical analysis.

## 3. Results

From January 2019 to December 2021, NAS delivered 2,798,348 patients aged 15 years or older (including 190,734 with COVID-19) and performed 1,557,388 COVID-19 screenings. The distribution of these activities based on gender and age are shown in Table 1 and Table 2.

Similarly, when comparing the distribution of gender and age regarding COVID-19 screenings and COVID-19-related ambulance deliveries (Table 2), the differences between 2020 and 2021 were always significant (*p* < 0.001).

Table 3 shows that the number of deliveries involving acute myocardial infarction significantly increased between the 2019/2020 (*p* < 0.001) and the 2019/2021 (*p* < 0.001) data. However, no significant change was observed between 2020 and 2021 (*p* = 0.815). Regarding haemorrhagic stroke, a similar pattern is shown but with a negative correlation. A significant decrease was detected when comparing the 2019/2020 (*p* < 0.001) and the 2019/2021 data (*p* < 0.001), but no significant difference was found when comparing the 2020 and 2021 data (*p* = 0.268). The deliveries of ischemic stroke cases consistently and significantly rose each year (*p* < 0.001). As the majority of stroke cases were recorded as ischemic, all stroke deliveries also significantly increased each year (*p* < 0.001). Overall non-COVID-19 deliveries significantly rose from 817,857 cases in 2019 to 841,427 cases in 2020 (*p* < 0.001) and continued to increase significantly to 948,330 cases in 2021 (*p* = 0.005). Both the number of COVID-19 screenings and COVID-19-related ambulance deliveries increased significantly from 2020 to 2021 (*p* < 0.001).

Figure 1A shows that the weekly average of deliveries involving acute myocardial infarction rose steadily during the three years and peaked at the beginning of the second COVID-19 wave. During this peak, the upper 95% standard deviation threshold was also breached. The R-squared value was only 0.073, partly because the number of deliveries between 2020 and 2021 did not differ significantly (see Table 3). Figure 1B shows that the increase in the weekly average of deliveries regarding stroke increased more considerably. The peak occurred during the third wave, during which the upper 95% standard deviation threshold was breached multiple times. With an R-squared value of 0.2197, the linear trend better matched the data points compared to acute myocardial infarction. The weekly moving average of overall non-COVID-19-related ambulance deliveries are shown in Figure 1C. The lower 95% standard deviation threshold was almost surpassed during the lockdown of the first COVID-19 wave, after which the weekly moving average of deliveries increased consistently. The peak occurred during the third wave, after which the deliveries did fall below the upper 95% standard deviation threshold. The R-squared value of the linear trend was 0.4008. Low degree of all the R-squared values for the weekly variations may have also played a role.

Figure 2A,B illustrate the weekly moving average of COVID-19 screenings and COVID-19-related ambulance deliveries, respectively. The intensity of these activities is consistent with the COVID-19 waves. In terms of acute myocardial infarction, the gender and age distribution did not change significantly over a three-year period (see Appendix A). For haemorrhagic stroke, only the age distribution changed significantly (*p* = 0.045) between 2019 and 2020, and for ischemic stroke a significant difference was observed with respect to age between 2019 and 2021 (*p* = 0.016). After merging the two types of strokes, the significant difference persisted for age when comparing 2019 and 2020 data (*p* = 0.015) and when comparing 2019 and 2021 data (*p* = 0.006). For all non-COVID-19-related deliveries, the distribution for both gender and age was significant across all three years (*p* < 0.05).

## 4. Discussion

In general, the rescue activities of the NAS showed a significant increase during the periods of the COVID-19 pandemic. Of the four pandemic waves that have taken place to date in Hungary, the third COVID-19 wave required the greatest effort from the NAS. AMI and stroke are primary issues from a public health point of view [24,25], so it is important for the healthcare system that the ambulance service prioritizes the organization and execution of such rescues [13,26]. A previous British study did not report increased STEMI and stroke rescue activity during the period of the COVID-19 epidemic, but only the effect of the first wave was analyzed in their study [16]. This research result was also confirmed by our study, as during the first wave of COVID, we did not observe a significant increase in the rescue activity tasks, possibly because the additional rescue tasks caused by COVID-19 may have been offset by the reduced rescue burden as a consequence of the lockdown (e.g., reduced number of injuries from motor vehicle collisions and sports-related activities) [27].

The NAS has several instruments at their disposal to perform at the highest possible level. One of the most often-used tools is business intelligence (BI). “BI is a technology-driven process for analyzing data and delivering actionable information that helps executives, managers and workers make informed business decisions” [28]. In Hungary, with the help of this system, the management of the NAS is able to monitor—at a national level on a daily basis—whether the number of rescue units is sufficient to perform the tasks at hand. As other studies have demonstrated, the appropriate use of BI serve as an effective tool for decision makers, even during a pandemic [29,30,31]. Therefore, it can be assumed that the NAS also benefited from using BI.

For healthcare providers around the world, in order to perform the extra tasks that arose during the COVID-19 pandemic in a professional and timely manner, additional human resources were needed. This brought about a change not only in the number of employees but also in their composition, in addition to the need to ensure that staff did not become infected [32,33,34,35]. For the NAS, this meant that medical and health science university students (e.g., physiotherapy and public health students) were also involved in the execution of tasks according to their knowledge and skills. Furthermore, a procedure for sampling required by general practitioners was established, for which the work of the Red Cross and volunteers was organized. Additionally, consultations were held several times a week with the institutions providing inpatient COVID-19 care with respect to the number of beds available.

A limitation of our study is that we were only able to define one year as a baseline before the onset of the COVID-19 pandemic. The number of haemorrhagic stroke cases appeared to be exiguous during our investigation because the ambulance service is not able to clearly separate cases of stroke, so the two types of strokes were combined in our analysis. However, the strength of our study is that we were able to examine 3 years of data provided directly by the domestic ambulance service. The authentic and professionally relevant data were ensured by the NAS and verified, with the help of BI, from the point of view of a medical professional. However, as the ambulance staff was responsible for the initial data recording, in cases of severe emergency there, was not enough time to enter all supplementary data, which might explain why approximately 10% of gender and approximately 1% of age data were not recorded. Finally, in the case of AMI and stroke, because the distribution did not differ significantly according to age and gender, we could rule out that the increase in the number of rescues was the result of a change in the composition of those who required ambulance delivery.

## 5. Conclusions

As a significant increase in ambulance deliveries was identified, the study results confirm that the emergence of a pandemic puts a heavy burden on emergency services, although the degree of variation may depend on the type of disease. The increase in activities coincided with the pandemic waves; however, our study does not provide an explanation of why the deliveries related to acute myocardial infarction and stroke responded differently. For example, a possible explanation for the fewer haemorrhagic stroke deliveries is that during the lockdown of the first wave, there were fever road accidents. However, without additional data, this cannot be confirmed or rejected. Thus, future studies should investigate this enigma. Furthermore, we recommend that future studies expand the timeframe and investigate how ambulance deliveries changed after the fourth wave of the COVID-19 pandemic, in addition to broadening the scope of the investigated diseases to determine which other types of deliveries are most affected by the pandemic. The aforementioned limitations must also be considered when interpreting the results of the present study, and future studies should address these limitations to provide more robust evidence. This can be achieved by including at least two years of baseline data for comparison and clearly separating the two types of strokes with diagnostic imaging.

Finally, emergency services must also prepare for the next global pandemic; to this end, it is important to rethink capacity organization and rescue activities. The Hungarian example highlights that in a pandemic situation, it can be beneficial to organize the emergency care of a country or a larger region under a single provider with a single decision-maker, and BI can provide assistance in monitoring critical indicators. Finally, we suggest that, similar to the NAS, the emergency services of various countries share their strategies by which they were able to cope with the increased burden during the pandemic.

## Figures and Tables

**Figure 1 healthcare-10-02331-f001:**
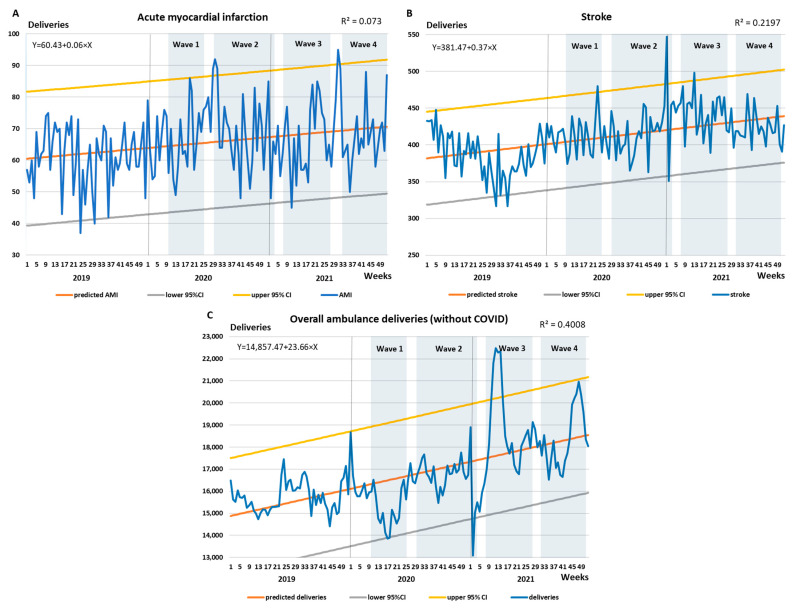
Weekly average of (**A**) acute myocardial infarction, (**B**) stroke, and (**C**) overall non-COVID-related deliveries by the NAS between 2019 and 2021. The upper and lower 95% confidence intervals are calculated based on the predicted values. The R-squared values of the linear trends are shown in the upper-right corners, and the equations of the regression lines are shown in the upper-left corners.

**Figure 2 healthcare-10-02331-f002:**
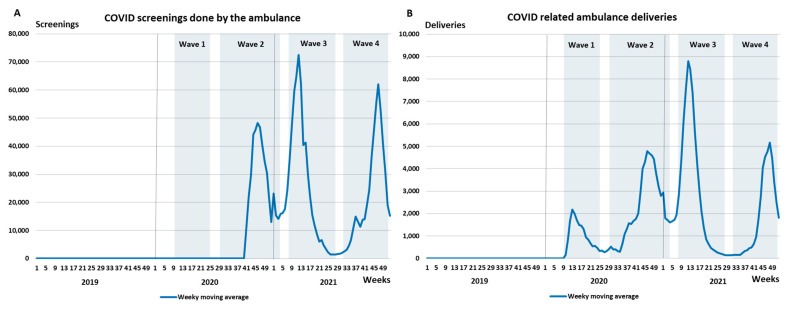
Weekly moving average of (**A**) COVID-19 screenings and (**B**) COVID-19-related deliveries by the NAS between 2019 and 2021.

**Table 1 healthcare-10-02331-t001:** Ambulance activity in Hungary, 2019–2021.

Variable	Ambulance Deliveries					
	2019		2020		2021	
	N	%	N	%	N	%
Activity related to:						
COVID-19 screenings	N.A.	N.A.	387,116	31.5%	1,168,890	55.2%
COVID-19 diagnosis	N.A.	N.A.	71,512	5.8%	118,508	5.6%
non-COVID-19 diagnosis	817,857	100%	769,915	62.7%	829,822	39.2%
Accompanied by:						
Acute myocardial infarction	3161	0.4%	3545	0.3%	3524	0.2%
Haemorrhagic stroke	362	0.0%	240	0.0%	219	0.0%
Ischemic stroke	20,017	2.4%	21,441	1.7%	22,581	1.1%
Stroke	20,379	2.5%	21,681	1.8%	22,800	1.1%
Gender						
male	316,618	38.7%	327,081	38.9%	372,144	39.2%
female	332,182	40.6%	340,076	40.4%	393,422	41.5%
missing	169,057	20.7%	174,270	20.7%	182,764	19.3%
Age						
15–64	424,215	51.9%	430,253	51.1%	482,786	50.9%
65+	350,751	42.9%	367,252	43.6%	424,655	44.8%
missing	42,891	5.2%	43,922	5.2%	40,889	4.3%

**Table 2 healthcare-10-02331-t002:** Number and distribution of COVID-19 screenings and COVID-19-related deliveries by the NAS between 2020 and 2021.

	2020		2021		2020/2021
	N	%	N	%	*p*-Value
COVID-19 screenings performed by the ambulance service					
Gender					
male	36,738	9.4%	495,871	42.5%	<0.001 *
female	55,563	14.2%	670,056	57.5%	
missing	299,155	76.4%	5	0.0%	
Age					
15–64	345,913	88.4%	1,009,088	86.6%	<0.001 *
65+	45,432	11.6%	156,811	13.4%	
missing	111	0.0%	33	0.0%	
COVID-19-related ambulance deliveries					
Gender					
male	31,840	44.1%	53,458	45.1%	<0.001 *
female	32,909	45.5%	55,594	46.9%	
missing	7501	10.4%	9432	8.0%	
Age					
15–64	33,546	46.4%	59,149	50.0%	<0.001 *
65+	37,748	52.3%	59,165	49.9%	
missing	956	1.3%	170	0.1%	

* Significant (*p* < 0.05).

**Table 3 healthcare-10-02331-t003:** Comparative analysis of COVID-19 screenings and various types of deliveries by the NAS between 2019 and 2021.

		Weekly Statistics			*p*-Value		
		2019	2020	2021	2019/2020	2019/2021	2020/2021
Acute myocardial infarction (ICD10: I21, I22)	N (yearly data)	3161	3545	3524	<0.001 *	<0.001 *	0.815
	Mean	60.8	68.2	67.8			
	SD	9.5	10.8	11.1			
	Median	61.5	69	66.5			
	IQR	12	16	12.5			
	Shapiro–Wilk	0.036	0.640	0.574			
Haemorrhagic stroke (ICD10: I60, I61, I62)	N (yearly data)	362	240	219	<0.001 *	<0.001 *	0.268
	Mean	7.0	4.6	4.2			
	SD	3.1	2.0	1.7			
	Median	6.5	4	4			
	IQR	5.25	3	2.25			
	Shapiro–Wilk	0.050	0.293	0.902			
Ischemic stroke (ICD10: I63, I64)	N (yearly data)	20,017	21,441	22,581	<0.001 *	<0.001 *	<0.001 *
	Mean	384.9	412.3	434.3			
	SD	30.8	25.2	32.1			
	Median	384.5	417.5	430			
	IQR	45.5	39.75	41.5			
	Shapiro–Wilk	0.593	0.526	0.068			
Stroke (ICD10: I60, I61, I62, I63, I64)	N (yearly data)	20,379	21,681	22,800	<0.001 *	<0.001 *	<0.001 *
	Mean	391.9	416.9	438.5			
	SD	31.7	25.5	31.8			
	Median	390	421.5	435			
	IQR	46.5	40	42			
	Shapiro–Wilk	0.571	0.626	0.074			
Overall ambulance deliveries (without COVID-19 deliveries)	N (yearly data)	817,857	841,427	948,330	<0.001 *	<0.001 *	0.005 *
	Mean	15,728.0	16,181.3	18,237.1			
	SD	670.9	1014.9	1882.5			
	Median	15,570.5	16,364.5	18,096			
	IQR	858.5	1071	1806			
	Shapiro–Wilk	0.140	0.203	0.168			
COVID-19 screenings performed by the ambulance service	N (yearly data)	0	387,116	1,168,890	N.A. **	N.A. **	<0.001 *
	Mean	N.A. **	7444.5	22,478.7			
	SD	N.A. **	15,028.5	19,958.9			
	Median	N.A. **	0	15,426.5			
	IQR	N.A. **	0.25	29,035.75			
	Shapiro–Wilk	N.A. **	<0.001	<0.001			
COVID-19-related ambulance deliveries	N (yearly data)	0	71,512	118,508	N.A. **	N.A. **	<0.001 *
	Mean	N.A. **	1375.2	2279.0			
	SD	N.A. **	1442.3	2380.6			
	Median	N.A. **	843	1683.5			
	IQR	N.A. **	1492	3258.25			
	Shapiro–Wilk	N.A. **	<0.001	<0.001			

* Significant (*p* < 0.05). ** Not applicable.

## Data Availability

The data presented in this study are available upon request from the corresponding author. The data are not publicly available at the request of the board members of the Hungarian National Ambulance Service.

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
