# Peer review of "The Emergency Performance of the Hungarian Ambulance Service during the COVID-19 Pandemic"

_healthcare, 2022, doi:10.3390/healthcare10112331_

Round 1

Reviewer 1 Report

Title: The emergency performance of the Hungarian ambulance service during the COVID-19 pandemic

This topic is interesting and timely. The authors could address a practical problem for the recent pandemic. However, there are some concerns to be resolved at this stage based on the following comments:

1-     Please polish the English of the manuscript. I found several mistakes.

2-     Numerical achievements of the study need to be highlighted in the abstract.

3-     Introduction section is ill-written. Problem significance, research questions/directions, a literature survey to highlight the contributions and structure of the remaining sections are missed.

4-     Give a better explanation of the methodology, maybe using a framework/flowchart. Please also discuss the pros and cons of your proposed methodology, especially against those given in the literature.

5-     Nice experiments are done. However, the main limitations of the study should be clearly discussed in the conclusion section, and then, a useful outlook to be given.

Author Response

Point 1: Please polish the English of the manuscript. I found several mistakes.

Response 1: We have asked one of our colleagues who speaks English fluently to carefully read the manuscript and check for grammatical errors. These changes are highlighted with ‘change tracked’ all across the manuscript. We hope that these alterations are adequate.

Point 2: Numerical achievements of the study need to be highlighted in the abstract.

Response 2: Due to the word limit of the abstract, we were unable to insert the key numerical results of the study. However, we agree, that the level of significance should be highlighted. Therefore, when we mention that “… a significant increase regarding all three medical conditions and overall deliveries.”, we now include that the p value for all the significant results is lower than 0.001 (see line 29).  

Point 3: Introduction section is ill-written. Problem significance, research questions/directions, a literature survey to highlight the contributions and structure of the remaining sections are missed.

Response 3: We agree that the original introduction overemphasized the Hungarian COVID-19 pandemic and the Hungarian National Ambulance Service, while gave little justification for the research and no research aim was given. Therefore, (1) we now mention the British study conducted on this topic, (2) we also highlight that no other studies investigated this question, and (3) we now finish the Introduction section with our research aim.  The new paragraph is written as follows: “The NAS already faced considerable challenges before the pandemic because the Hungarian health indicators are among the worst in the European Union [13-15]. Therefore, it is imperative to understand the degree the COVID-19 wave increased the workload of the “first responders” in healthcare. Despite the importance of the topic only one research investigated how emergency ambulance deliveries within the United Kingdom were affected by the COVID pandemic [16]. Since this study was conducted relatively early on during the pandemic, the researchers could only investigate the first COVID wave, thus, there is no information regarding the impact of the following, more severe waves, which could be useful for those organizing emergency ambulance services. Although other articles discussed the increased burden on ambulance services during the COVID-19 pandemic, these published papers mainly focused on how to identify COVID-19 infections and other factors related to medical technology [17-20]. Due to the lack of knowledge and the high possibility that in the not-so-distant future a new global pandemic could emerge, it is important to investigate this question thoroughly. Therefore, the primary aim of this research was to examine the performance of the NAS during the first four waves of the pandemic.”

Point 4: Give a better explanation of the methodology, maybe using a framework/flowchart. Please also discuss the pros and cons of your proposed methodology, especially against those given in the literature.

Response 4: In cases where a justification for the applied methodology is warranted, we now make short explanations:

(1) For the reason to investigate the two types of strokes together and independently from one another is now explained as: “Because the initial analysis revealed that the cases of hemorrhagic stroke were very low, and that the ambulance staff did not have the necessary tools to accurately identify the type of stroke during the delivery, the decision was made to merge the two types of strokes into one data. However, by doing so, relevant information might be lost, thus, the main statistical analyses were performed for both stroke types as well.”

(2) We now mention where the idea came from to use the 95% standard deviation: “In addition, we calculated the upper and lower 95% standard deviations using the 2019 year as a baseline reference. This approach was adapted from the British study investigating emergency ambulance deliveries during the first COVID wave for STEMI (ST-elevation myocardial infarction) and stroke [16]. By doing so, the goal was to explore which weeks had considerably more or less deliveries.”

(3) We also make an argument why we have opted to use the reproduction rate to calculate the beginning and ending of the four COVID waves: “There is an ongoing debate on how to define the start and the end of any given COVID wave [21,22]. Two methods were considered by our research team. However, we determined that adapting the method used in Chile might have its limitations in Hungary, as this approach assumes that the two countries have similar population density [21]. Therefore, we opted to use the method that considers the reproduction rate (R) of the virus [22].”

Point 5: Nice experiments are done. However, the main limitations of the study should be clearly discussed in the conclusion section, and then, a useful outlook to be given.

Response 5: We agree that it is important that the reader should be aware while reading the Conclusions section that the limitations of the study should be considered before deriving a firm conclusion. Therefore, we have altered the first sentence within the Conclusions section as follows: “The study confirms that the emergence of an epidemic puts a heavy burden on emergency services, although the degree of variation may vary depending on the type of disease and the aforementioned limitations should be considered when interpreting the results.”   

Reviewer 2 Report

I read the manuscript submitted for review with real interest. The data reported are truly significant and of great impact for readers.

Apart from some minor inaccuracies (Klara Biro is indicated as the corresponding author, but the asterisk is on Mate Sandor Deak; in mat and meth, line 67 no a uniform, but an uniform).

Tables 1 and 2 perplex me: what does no data mean with regard to gender and age?

Author Response

Point 1: Apart from some minor inaccuracies (Klara Biro is indicated as the corresponding author, but the asterisk is on Mate Sandor Deak; in mat and meth, line 67 no a uniform, but an uniform).

Response 1: Máté Sándor Deák is the corresponding author of this manuscript. During the finalization of the manuscript, we forgot to alter the e-mail address of the corresponding author. This is now corrected. Also, we have asked one of our colleagues who speaks English fluently to carefully read the manuscript and check for grammatical errors. All of these changes are highlighted with ‘change tracked’ all across the manuscript.

Point 2: Tables 1 and 2 perplex me: what does no data mean with regard to gender and age?

Response 2: Because the staff of the ambulance had to record these data, it is possible that in severe cases there was no time to record gender and age. This is indeed an important limitation of the dataset used. Thus, we have added the following two sentences in the last paragraph of the Discussion section: “However, as the ambulance staff was responsible for the initial data recording, in cases of severe emergency there was not enough time to enter all supplementary data. This might explain why around 10% of gender and around 1% of age were not recorded.”  

Reviewer 3 Report

The emergency performance of the Hungarian ambulance service during the  COVID-19 pandemic. 

This interesting topic is deeply investigated (Hungarian National Ambulance Service implemented data during the four wawes of the Covid 19 pandemic), and well discussed by the Authors in this article.   

Authors observed:  Soon after the appearance of this novel virus, it caused a considerable burden on the health care systems globally. Please consider in this regard also, aspects of health care responsibility (The medico-legal implications in medical malpractice claims during Covid-19 pandemic: Increase or trend reversal? Bilotta, C., Zerbo, S., Perrone, G., Malta, G., Argo, A. The Medico-legal  Journal, 2020, 88(1), pp. 35–37). 

The Authors correctly affirmed: Due to the lack of knowledge and the high possibility that in the not-so-distant future a new global pandemic could emerge again, it is important to investigate this question thoroughly.

It is the responsibility of science to indicate preventable measures for the near future in light of previous pandemic events (Please consider in this regard also: COVID-19 Pandemic: New Prevention and Protection Measures. Cirrincione, L.Plescia, F.Ledda, C., ...Vinnikov, D.Cannizzaro, E. Sustainability (Switzerland), 2022, 14(8), 4766.)

4. Discussion: With regard to BI (business intelligence), it should be clarified to the Authors the meaning of this term, as is not familiar to most readers.  

Author Response

Point 1: Authors observed:  Soon after the appearance of this novel virus, it caused a considerable burden on the health care systems globally. Please consider in this regard also, aspects of health care responsibility (The medico-legal implications in medical malpractice claims during Covid-19 pandemic: Increase or trend reversal? Bilotta, C., Zerbo, S., Perrone, G., Malta, G., Argo, A. The Medico-legal  Journal, 2020, 88(1), pp. 35–37).

Response 1: We have checked the recommended article and believe that this does strengthen the overall claim at the beginning of the manuscript, that the COVID-19 pandemic is an unparalleled challenge for the health sector. Therefore, we have supplemented the list of relevant articles with your recommendation.

Point 2: The Authors correctly affirmed: Due to the lack of knowledge and the high possibility that in the not-so-distant future a new global pandemic could emerge again, it is important to investigate this question thoroughly.

Response 2: We would like to thank you for this remark.

Point 3: It is the responsibility of science to indicate preventable measures for the near future in light of previous pandemic events (Please consider in this regard also: COVID-19 Pandemic: New Prevention and Protection Measures. Cirrincione, L., Plescia, F., Ledda, C., ...Vinnikov, D., Cannizzaro, E. Sustainability (Switzerland), 2022, 14(8), 4766.)

Response 3: We have checked the recommended article and found it relevant within the Discussion section. Therefore, we altered the beginning of the third paragraph within the Discussion section as follows: “For healthcare providers around the world in order to perform in a professional and timely manner the extra tasks that occurred during the COVID pandemic, additional human resources were needed. This brought about a change not only in the number of employees, but also in their composition, while also ensuring that the staff do not become infected [32-35].”

Reviewer 3.4: Discussion: With regard to BI (business intelligence), it should be clarified to the Authors the meaning of this term, as is not familiar to most readers. 

Response 3.4: We agree that we should have clarified what business intelligence means, as it is not a commonly used terminology within healthcare. Therefore, we have added an explanation within the second paragraph of the Discussion section as follows: “The NAS has several instruments at their disposal to be able to perform at the highest possible level. One of the most often used tools is Business Intelligence (BI). “BI is a technology-driven process for analyzing data and delivering actionable information that helps executives, managers and workers make informed business decisions” [28].”

Round 2

Reviewer 1 Report

Good work. The authors tried to address the comments. However, the conclusion section should be enriched by providing more explanations on the main findings, limitations of your study as well as a good outlook for future studies.

Author Response

Point 1: Good work. The authors tried to address the comments. However, the conclusion section should be enriched by providing more explanations on the main findings, limitations of your study as well as a good outlook for future studies.

Response 1: We would like to apologize for poorly rewriting the conclusion. Based on your suggestion we have expanded this section. The alterations were made as follows: “As significant increase in ambulance deliveries were identified, the study confirms that the emergence of an epidemic puts a heavy burden on emergency services, although the degree of variation may depend on the type of disease. The increase in activities coincided with the pandemic waves; however, our study does not give an explanation on why the deliveries related to acute myocardial infarction and stroke changed differently. Thus, future studies should investigate this enigma. Furthermore, we recommend that these researches expand the timeframe and investigate how ambulance deliveries changed after the fourth wave of the COVID pandemic, while also broadening the scope of the investigated diseases to see which other type of deliveries are most affected by the pandemic. Also, the aforementioned limitations must be considered when interpreting the results and future studies should tackle these to provide more robust evidence. This can be achieved by having at least two years of baseline for comparison and clearly separating the two types of strokes with imaging diagnostics.” Finally, some minor typos were also corrected using ‘Track Changes’